# Description of the First Registered Case of Lopes–Maciel–Rodan Syndrome in Russia

**DOI:** 10.3390/ijms232012437

**Published:** 2022-10-18

**Authors:** Yuliya S. Koshevaya, Aleksey V. Kusakin, Natalia V. Buchinskaia, Valentina V. Pechnikova, Elena A. Serebryakova, Alexander L. Koroteev, Andrey S. Glotov, Oleg S. Glotov

**Affiliations:** 1Saint-Petersburg State Medical Diagnostic Center (Genetic Medical Center), 353912 St. Petersburg, Russia; 2CerbaLab Ltd., 199106 St. Petersburg, Russia; 3Pediatric Research and Clinical Center for Infectious Diseases, 197022 St. Petersburg, Russia; 4Applied Genomics Laboratory, SCAMT Institute, ITMO University, 197101 St. Petersburg, Russia; 5Avtsyn Research Institute of Human Morphology of Federal State Budgetary Scientific Institution “Petrovsky National Research Centre of Surgery”, 117418 Moscow, Russia; 6Department of Genomic Medicine, D.O.Ott Research Institute of Obstetrics, Gynaecology and Reproductology, 199034 St. Petersburg, Russia

**Keywords:** Lopes–Maciel–Rodan syndrome, LOMARS, clinical forms of LOMARS syndrome, huntingtin, *HTT* gene, SNP

## Abstract

Lopes–Maciel–Rodan syndrome (LOMARS) is an extremely rare disorder, with only a few cases reported worldwide. LOMARS is caused by a compound heterozygous mutation in the *HTT* gene. Little is known about LOMARS pathogenesis and clinical manifestations. Whole exome sequencing (WES) was performed to achieve a definitive molecular diagnosis of the disorder. All NGS-identified variants underwent the Sanger confirmation. In addition, a literature review on genetic variations in the *HTT* gene was conducted. The paper reports a case of LOMARS in a pediatric patient in Russia. A preterm girl of non-consanguineous parents demonstrated severe psychomotor developmental delays in her first 12 months. By the age of 6 years, she failed to develop speech but was able to understand everyday phrases and perform simple commands. Autism-like behaviors, stereotypies, and bruxism were noted during the examination. WES revealed two undescribed variants of unknown clinical significance in the *HTT* gene, presumably associated with the patient’s phenotype (c.2350C>T and c.8440C>A). Medical re-examination of parents revealed that the patient inherited these variants from her father and mother. Lopes–Maciel–Rodan syndrome was diagnosed based on overlapping clinical findings and the follow-up genetic examination of parents. Our finding expands the number of reported LOMARS cases and provides new insights into the genetic basis of the disease.

## 1. Introduction

The *HTT* gene (OMIM 613004) is located on the short arm of chromosome 4 and encodes the nuclear protein huntingtin, expressed in various organs and tissues, with the highest expression levels observed in tissues of the nervous system. Huntingtin also interacts with hundreds of other proteins [1,2]. Toxic gain-of-function (GoF) expansion of the unstable CAG repeat on the *HTT* gene causes Huntington’s disease–a neurodegenerative disorder with the autosomal dominant mode of inheritance [3]. Motor, mental, cognitive, and other clinical symptoms can start in adolescence and would progress exacerbate in adulthood [4]. There are currently no available treatments that slow or prevent the manifestation of Huntington’s disease in affected patients. Potential disease-modifying therapies are aimed at suppressing the expression of either total or allele-specific mutant huntingtin and are currently undergoing clinical trials [5,6].

However, the correlation between the gain-of-function mechanism and normal huntingtin function or loss of function (LoF) remains unclear.

A remarkable study of the role of huntingtin through LoF mutations was performed in mice. Heterozygous mice with single-allele LoF mutations manifested viability, fertility, and normal phenotypes. In contrast, homozygous mice and mice with compound heterozygous LoF mutations showed several variations in developmental phenotypes, including death at different stages (fetal, postnatal, or adulthood) [7,8].

Little is known about the huntingtin function in humans, with a single reported family case of a balanced translocation [4,9] terminating at intron 40 of the *HTT* gene. The allele showed no huntingtin expression and did not produce relevant clinical manifestations in two heterozygous family members who carried a singly intact *HTT* gene allele [10,11].

Based on the overall evidence, we can assume that a single intact *HTT* allele is sufficient to sustain normal development.

By now, investigators have described a few family cases, carrying two putative compound heterozygous LoF mutations in the *HTT* gene, responsible for the development of Lopes–Maciel–Rodan syndrome (a rare congenital disease with Rett-like neurological symptoms). So far, LOMARS syndrome is known to be associated with a variety of symptoms: limb spasticity, decreased muscle tone (hypotonia), stereotyped arm movements, dystonia, ataxia, epilepsy, myopia, bruxism, etc. [9,12].

## 2. Results

### 2.1. Early Disease Manifestation and Clinical History of the Proband

The child is 7 years and 11 months old. Her mother’s primary complaints are lack of speech, mental retardation, impaired gait, and movement coordination. The patient is 132 cm high and weighs 35 kg. Upon examination, the patient presented with automatic behavior, stereotypy, and bruxism. The patient also had characteristic dysmorphic facial features, including a long-shaped face, a high symmetrical forehead, ocular hypotelorism, intermittent divergent strabismus, a thin upper lip, and a nearly permanently open mouth. Her skin appeared pink and clean, and her hair appeared thick and blond. Heart tones were rhythmic and of normal intensity. Palpation of the abdomen revealed no hepatosplenomegaly. External genitalia was of the female type. The patient’s gait was altered, with notable ataxia, a knee-bent walking pattern, and lower limb valgus deformity. The patient exhibited right-side hemiparetic muscle weakness. Tendon reflex was brisk due to an increase in the expansion of reflexogenic zones. Cognitive functions appeared reduced. The patient was able to perform simple commands and requests. The patient’s Speech ability was limited to separate syllables and sounds, showing no phonation or swallowing impairments. The patient spontaneously used gestures to communicate desires.

Anamnesis morbi: A premature baby girl was delivered by a G5P2 mother. The baby suffered hypoxia during delivery, ranking 3–6 by the Apgar score. The condition at birth was severe (for the next 13 days, the patient stayed in the ICU on artificial respiration). The discharge diagnosis was defined as hypoxic and an ischemic central nervous system injury. Before reaching the age of 12 months, the patient demonstrated 8–10 weeks of developmental retardation. At the age of 7 months, the mother noticed a divergent squint (strabismus). At the age of 8 months, the child was diagnosed with infantile cerebral palsy based on a neurological examination. Between the ages of 9 and 24 months, the patient underwent intensive rehabilitation treatment, including reflex locomotion (also referred to as the Vojta therapy), massage, therapeutic exercises, and osteopathy. The rehabilitation efforts eventually produced slow improvements in psychomotor development. At the age of 10 months, the girl started to acquire ‘commando’ crawling; at the age of 15 months, she began to crawl on her hands and knees; by the age of 18 months, she learned to stand on her feet with support. She said her first words at the age of 20 months. She first walked independently at the age of 30 months.

The child, assisted by a therapist and psychologist, learned to read and pronounce simple syllables at 4 to 5 years of age. At the age of 5 years, the patient underwent her first genetic examination. A 46, the XX (normal female) karyotype was revealed and the Niemann–Pick type C disease (manifesting itself as ataxia) was excluded based on biochemical findings. Tandem mass spectrometry was performed to exclude versatile hereditary metabolic diseases. The examination eventually did not produce any evidence of disease. By the age of 6 years, speech skills were still not developed with persisting hemiparesis and ataxia; the patient was able to understand everyday phrases and execute basic commands.

Awake and sleep video EEG monitoring recorded epileptiform activity, morphologically resembling benign focal epileptiform discharges of childhood (BFEDCs). MRI revealed no structural abnormalities without evidence of perinatal hypoxic brain damage. Thus, genetic pathology was suspected, suggesting atypical Rett syndrome. A further neurological examination identified an atypical form of self-limited epilepsy with centrotemporal spikes (SeLECTS), combined with mild intellectual disability. Awake EEG monitoring demonstrated altered bioelectrical activity due to moderate diffuse cerebral dysfunctions associated with the dysregulation processes. Sleep EEG produced properly shaped waves with no evidence of abnormal sleep architecture. No typical epileptiform activity was registered; an ophthalmic examination detected mild hypermetropia. The current neurological diagnosis is defined as infantile cerebral palsy, spastic diplegia of moderate severity, delayed mental and speech development, EEG patterns showing pathological activity, and divergent strabismus.

### 2.2. Whole Exome Sequencing and Sanger Sequencing

Whole exome sequencing revealed two c.8440C>A (chr4:4:3233337 (GRCh38), chr4:3235064 (GRCh37)) and c.2350C>T (chr4:3132675 (GRCh38), chr4:3134402 (GRCh37)) mutations in the *HTT* gene. Sanger sequencing was performed to confirm the variants. Parental re-examination showed that the c.2350C>T variant was passed on by the father (Figure 1), whereas c.8440C>A was inherited from the mother (Figure 2), thus confirming compound heterozygous variants in the proband.

The paternally inherited variant c.2350C>T is responsible for the arginine to cysteine substitution at codon 784 (p.Arg784Cys) and is present in gnomAD and ExAC control datasets at extremely low frequencies, i.e., <0.01% (7 alleles in 249,572 and 4 alleles in 120,886, respectively). This genetic variant is never-homozygous. VARITY and MutationTaster predictive in silico tools indicate that this genetic variant causes structural and functional disruptions in proteins [13,14].

The maternally inherited variant c.8440C>A is responsible for leucine to methionine substitution at codon 2814 (p.Leu2814Met) and is absent in the control samples. MutationTaster predictive in silico software indicates that this genetic variant causes structural and functional disruptions in proteins [14].

The aggregate data suggest that the identified c.2350C>T and c.8440C>A variants, which are of unknown clinical significance, are relevant to the proband phenotype (ACMG, Guidelines for the interpretation of massively parallel sequencing variants [15,16]).

## 3. Discussion

Lopes–Maciel–Rodan syndrome (LOMARS) is an extremely rare disorder, with only a few cases reported worldwide. LOMARS is caused by a compound heterozygous mutation in the *HTT* gene and manifests as a wide range of neuropsychiatric symptoms that obscure genetic diagnosis.

This paper describes the first case of LOMARS syndrome registered in the Russian Federation. The patient presented with a rare Rett-like syndrome and revealed two putative compound heterozygous LoF mutations in the *HTT* gene, which were inherited from both parents. The patient exhibited a few LOMARS symptoms, including limb spasticity, stereotyped arm movements, dystonia, ataxia, and bruxism. In addition, the patient’s medical record contained evidence of atypical self-limited epilepsy with centrotemporal spikes (SeLECTS). Rodan L. et al. and Lopes F. et al. also reported epilepsy and/or seizure activity in some patients, which may be regarded as a diagnostic criterion for LOMARS [9,12]. Other LOMARS manifestations may include refractive errors, as high myopia is reported in Rodan L. et al. [12]. Our study reports evidence of mild hypermetropia as well.

Lopes F. et al. refer to two heterozygous missense mutations c.2108C>T (p.Pro703Leu) and c.3779C>T (p.Thr1260Met) in exons 16 and 29, respectively, as possible causes of LOMARS (Table 1) [9]. In addition, Rodan L. et al. describe the c.4463+1G>A mutation in intron 34 and the c.8156T>A (p.Leu2719Gln) mutation in exon 60 (Table 1) [12]. However, neither of these papers report the two heterozygous c.2350C>T (p.Arg784Cys) and c.8440C>A (p.Leu2814Met) mutations in exon 17 and exon 61 of the *HTT* gene, respectively. The effects of these mutations are yet to be determined (Table 1).

Given the presented clinical and molecular genetic data, we recommend considering the probability of LOMARS in children with Rett-like neurological symptoms and perform molecular genetic testing to search for putative LoF mutations in the *HTT* gene sequence.

Elaborate diagnostic assessment and adequate interpretation of findings are pivotal for the rapid accurate diagnosis of inherited neuropsychiatric diseases.

The present study is an attempt to raise awareness among physicians regarding this rare condition and facilitate its diagnosis and molecular genetic confirmation in the future.

## 4. Materials and Methods

### 4.1. Study Participants and Samples Preparations

Since the age of 6 years, the patient was followed-up at a local public medical diagnostic center in St. Petersburg, Russia. Our study was approved by the Academic Review Board of the D.O. Ott Research Institute of Obstetrics Gynecology and Reproductology (St. Petersburg, Russia), resolution no. 113, dated 18 November 2021. For the purpose of research, all the patients signed written informed consent. The study was performed in accordance with the Declaration of Helsinki.

Blood samples from all family members were collected. Blood samples were stored using complex laboratory facilities for large-scale studies #3076082 “Human Reproductive Health”. For all blood samples, DNA was isolated through the phenol extraction method. Quantus Fluorometer^TM^ and QuantiFluor R dsDNA System (Promega Corporation, Madison, WI, USA) were used to determine DNA concentrations. DNA electrophoresis in 0.6% agarose gel in sodium borate (SB) buffer was used to assess DNA integrity.

### 4.2. Whole Exome Sequencing

Whole exome sequencing was performed using the Illumina sequencing platform. We prepared a gDNA library with 500 ng of gDNA sheared to ∼300 bp using a Covaris S2 focused-ultrasonicator, according to the manufacturer’s protocols. Upfront DNA fragmentation and DNA library preparation were conducted using the KAPA HyperPrep Kit (Roche Applied Science, Pleasanton, CA, USA) according to the manufacturer’s protocol. Library quantitation was performed using a Quantus Fluorometer with QuantiFluor^®^ dsDNA System kit (Promega, Madison, WI, USA). A High Sensitivity DNA assay with gel electrophoresis on the 2100 Bioanalyzer System (Agilent Technologies, Santa Clara, CA, USA) was used for DNA sizing accuracy and quality control (between 300 and 400 bp).

To generate exome-enriched DNA libraries, the HyperCap Target Enrichment kit and KAPA HyperExome Probes (Roche Applied Science, Pleasanton, CA, USA) were used according to the manufacturer’s protocol.

According to the Illumina protocol, paired-end reads of 2 × 100 bp were loaded on the HiSeq 2500 System for further sequencing.

### 4.3. Bioinformatics Data Analysis

Sequencing reads were aligned to the b37 human reference genome assembly using bwa mem v. 0.7.15-r1140 [17]. Alignment files were pre-processed, and variants were called using the Genome Analysis ToolKit (GATK) v. 3.5.0 [18]. Variant cohort calling and genotyping were performed using DeepVariant v0.10.0 [19]. Variants were filtered using Variant Quality Score Recalibration (VQSR) with truth sensitivity thresholds of 99.9 (for SNPs) and 99.0 (for indels). Filtered variants were annotated using the Ensembl Variant Effect Predictor (VEP) v103.1 by the following reference databases: 1000 Genomes, phase 3 [20]; exome aggregation consortium [21]; in-house Russian exome allele frequencies [22,23], as well as NCBI ClinVar and dbNSFP v 2.9 [24]. Custom software was used for enhanced variant interpretation.

### 4.4. Sanger Sequencing

In order to validate the identified candidate variants, Sanger sequencing of the proband and parental DNA was conducted. For this purpose, a custom pair of primers were designed (GGCCTGCTGTGTAGTCTCTTCTG (forward) and ACCTGGGTTCCCACTGATACCT (reverse)) for c.8440C>A and (TAGTAGGGAATTGTTCCATGGC (forward) and GAAGATTATGCTCACCCTCACAG (reverse) for c.2350C>T) to amplify the corresponding DNA fragment and run a sequencing analysis using the ABI 3500X platform. Visual inspection of the sequencing chromatograms confirmed the presence of the mutant sequence.

## Figures and Tables

**Figure 1 ijms-23-12437-f001:**
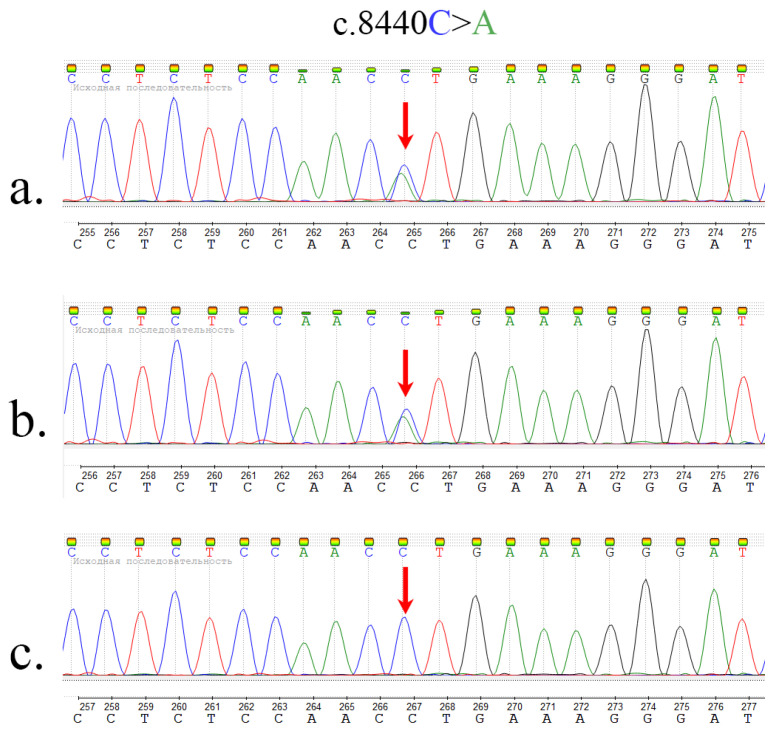
Sanger sequencing results: visual inspection of the sequencing chromatograms (forward primer for c.8440C>A). (**a**) Proband result. (**b**) Mother result. (**c**) Father result.

**Figure 2 ijms-23-12437-f002:**
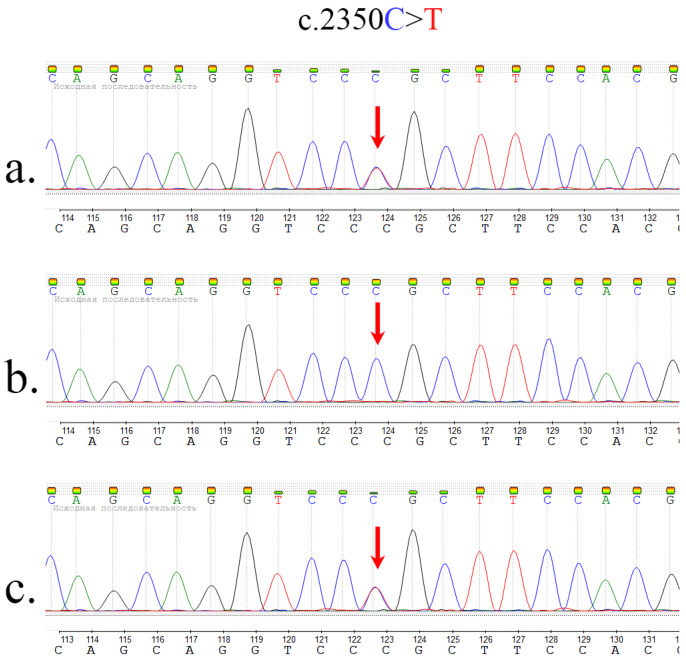
Sanger Sequencing results: visual inspection of the sequencing chromatograms (Forward primer for c.2350C>T). (**a**) Proband result. (**b**) Mother result. (**c**) Father result.

**Table 1 ijms-23-12437-t001:** LOMARS-causing mutations in *HTT*.

Reference	Position	Exon	cDNA Variant (aa)	Allele Frequency *
Lopes F. et al. [9]	chr4:3133374 C>T (GRCh37/hg19)	16	c.2108C>T p.Pro703Leu rs768047421	<0.01%
Lopes F. et al. [9]	chr4:3162034 C>T (GRCh37/hg19)	29	c.3779C>T p.Thr1260Met rs34315806	0.83%
Rodan L. et al. [12]	chr4:3177388 G>A (GRCh38)	34 (intron)	c.4463+1G>A rs1060505027	N/A
Rodan L. et al. [12]	chr4:3229927 T>A (GRCh38)	60	c.8156T>A p.Leu2719Gln rs1060505028	N/A
Current study	chr4:3134402 C>T	17	c.2350C>T p.Arg784Cys rs375919976	<0.01%
Current study	chr4:3235064 C>A	61	c.8440C>A p.Leu2814Met	N/A

* Allele frequencies according to the Genome Aggregation Database.

## Data Availability

Raw data are available from the corresponding author upon request.

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
