# Peer review of "Description of the First Registered Case of Lopes–Maciel–Rodan Syndrome in Russia"

_ijms, 2022, doi:10.3390/ijms232012437_

Round 1

Reviewer 1 Report

This article is a case report on a Russian patient with a very rare disorder known as LOMARS. LOMARS is caused by loss-of-function mutations in the huntingtin gene, the same that causes Huntington´s disease. It is an interesting description, because very little is known about this disorder, and about the biological functions of huntingtin. Moreover, this disorder is very difficult to diagnose due to clinical similiarities to others, such as Rett syndrome. The fact that is the only case described in Russia makes it more interesting. Although the disorder is rare, a rigorous record of LOMARS cases can contribute to understanding the role of huntingtin in neural development.

The paper is relatively well structured and provides a detailed description of the clinical issues of the patient since she was born. However, for unknown reasons, the authors name "huntingtin" as "gentingtin", a major naming mistake that must be corrected. This mistake in a key word of the manuscript does not give much confidence to this reviewer about the scientific rigor of the authors. The article would also benefit from proof-reading or professional editing by a native speaker, as English and writing style could be improved. For example, the authors combine past and present sentences in various paragraphs, lacking the necessary tense consistency.

In line 59, the authors say "There were no pathological signs." in the context of describing many pathological signs. This sentence is either wrong or ambiguous, and it is unclear what does it refer to: tendon reflexes, other neurological symptoms, cognitive functions???

In line 133, what is hyperopia? Do the authors mean hypermetropia or farsightedness or longsightedness? I am not a physician, and I cannot evaluate medical terms, but this sort of mistake does not make me feel confident about the scientific rigor of the authors. So, please, make sure that EVERY medical term in the manuscript is correct. Giving it to a professional scientific writer can help to detect and correct these mistakes.

Tables I and II are not really necessary, as the information is in the main text. If you want to maintain them, you can fuse them in a single one.

Line 158: "from family of three" I understand what it means, but I don´t think it is correct.

The methods section has several grammar mistakes. Again, proofreading by professional scientific writers is a must.

Author Response

Point 1: In line 59, the authors say "There were no pathological signs." in the context of describing many pathological signs. This sentence is either wrong or ambiguous, and it is unclear what does it refer to: tendon reflexes, other neurological symptoms, cognitive functions???

Response 1: This sentence is referring to the previous one which is describing tendon reflexes. I agree it is misleading and can be deleted.

Point 2: In line 133, what is hyperopia? Do the authors mean hypermetropia or farsightedness or longsightedness? I am not a physician, and I cannot evaluate medical terms, but this sort of mistake does not make me feel confident about the scientific rigor of the authors. So, please, make sure that EVERY medical term in the manuscript is correct. Giving it to a professional scientific writer can help to detect and correct these mistakes.

Response 2: As I know hyperopia is a synonym of hypermetropia.

Point 3: proofreading by professional scientific writers is a must.

Response 3: I agree. Significant revisions have been made, and the edited article is ready for uploading.

Reviewer 2 Report

As Lopes-Maciel-Rodan syndrome (LOMARS) is an extremely rare disorder about 100  cases reported worldwide. The manuscript extended our knowledge of LOMARS. Here, the authors describe the first case of LOMARS in Russia caused by compound heterozygous mutation in the HTT gene. Each mutation was detected respectively in proband's mother and father. Both variants are predicted  as disrupting protein structure and function. The article is written in good language and will be interesting to physicians as well as genetics. 

Author Response

Point 1: English language and style are fine/minor spell check required.

Response 1: A professional translator has revised the article and it is ready for uploading.

Round 2

Reviewer 1 Report

The authors have address my main concerns.